# Analysis of CVC1302-Mediated Enhancement of Monocyte Recruitment in Inducing Immune Responses

**DOI:** 10.3390/vaccines12010086

**Published:** 2024-01-15

**Authors:** Haiyan Lu, Xiaoming Yu, Liting Hou, Yuanpeng Zhang, Lan Li, Xuwen Qiao, Haiwei Cheng, Luping Du, Jin Chen, Qisheng Zheng, Jibo Hou

**Affiliations:** 1Institute of Veterinary Immunology & Engineering, Jiangsu Academy of Agricultural Sciences, Nanjing 210014, China; 2National Research Center of Engineering and Technology for Veterinary Biologicals, Jiangsu Academy of Agricultural Science, Nanjing 210014, China; 3Jiangsu Key Laboratory for Food Quality and Safety-State Key Laboratory Cultivation Base, Ministry of Science and Technology, Nanjing 210014, China; 4Guo Tai (Taizhou) Center of Technology Innovation for Veterinary Biologicals, Taizhou 210014, China; 5College of Veterinary Medicine, Nanjing Agricultural University, Nanjing 210095, China

**Keywords:** CVC1302, monocyte, CXCL9, CXCL10, antigen

## Abstract

Monocytes (Mos) are believed to play important roles during the generation of immune response. In our previous study, CVC1302, a complex of PRRs agonists, was demonstrated to recruit Mo into lymph nodes (LNs) in order to present antigen and secret chemokines (CXCL9 and CXCL10), which attracted antigen-specific CD4^+^ T cells. As it is known that Mos in mice are divided into two main Mo subsets (Ly6C^+^ Mo and Ly6C^−^ Mo), we aimed to clarify the CVC1302-recruiting Mo subset and functions in the establishment of immunity. In this study, we found that CVC1302 attracted both Ly6C^+^ Mo and Ly6C^−^ Mo into draining LNs, which infiltrated from different origins, injection muscles and high endothelial venule (HEV), respectively. We also found that the numbers of OVA^+^ Ly6C^+^ Mo in the draining LNs were significantly higher compared with OVA^+^ Ly6C^−^ Mo. However, the levels of CXCL9 and CXCL10 produced by Ly6C^−^ Mo were significantly higher than Ly6C^+^ Mo, which plays important roles in attracting antigen-specific CD4^+^ T cells. Under the analysis of their functions in initiating immune responses, we found that the ability of the Ly6C^+^ monocyte was mainly capturing and presenting antigens, otherwise; the ability of the Ly6C^−^ monocyte was mainly secreting CXCL9 and CXCL10, which attracted antigen-specific CD4^+^ T cells through CXCR3. These results will provide new insights into the development of new immunopotentiators and vaccines.

## 1. Introduction

Mos account for ~10% of nucleated blood cells, and play important roles in initiating adaptive immunity against infection and injury [1,2,3,4]. Mos are conventionally grouped into two main different subpopulations based on cell surface molecules and functions [5]. In mice, these subpopulations correspond to Ly6C^+^ CCR2^hi^ CX3CR1^lo^ classical Mo and Ly6C^−^ CCR2^lo^ CX3CR1^hi^ non-classical Mo and, in humans, the equivalent subpopulations are CD14^+^ CD16^−^ Mo and CD14^+^ CD16^+^ Mo [6,7].

Varied levels of the surface markers on Mo subsets reflect their functions. For example, Ly6C^+^ Mos express high levels of both CD64 and CD163, which enable its playing of important roles in innate immune response. Ly6C^+^ Mos also highly express CCR2, which promotes their migratory capacity to inflamed tissue, where they differentiate into Mo-derived dendritic cells (mo − DC) and macrophages, and produce inflammatory cytokines and reactive oxygen species in order to stimulate effector T cells [8,9,10]. Ly6C^−^ Mos have high levels of 6-sulfo LacNAc (SLAN), siglec10 and TNFR2, demonstrating their potential role in inflammation. Ly6C^−^ Mos patrol along the vascular endothelium and trans-endothelial migration dependent on high levels of CX3C motif chemokine receptor (CX3CR1) and CD11c [11,12].

In response to TLR agonists, Ly6C^+^ Mos were the most prominent producers in the secretion of IL-6, IL-8, CCL2 and CCL3, compared with Ly6C^−^ Mos [8,13]. Ly6C^−^ Mos secreted IFN-α in response to intracellular TLR3 stimulation, and secreted TNFα, IL-1β, and CCL3 in response to viral nucleic acids and immune complexes. Furthermore, Ly6C^+^ Mos differentiate into mo-DC under the incubation of IL-4 and granulocyte-macrophage colony-stimulating factor (GM-CSF), Ly6C^−^ Mo do not [8]. None of the Mo subset differentiate into plasmacytoid DC (pDC) under the incubation of IL-3 and FMS-like tyrosine kinase 3 ligand (Flt3L). All of the Mo subsets differentiate into macrophages under the incubation of GM-CSF or M-CSF [14,15].

In our previous study, we found that CVC1302 recruited significantly higher numbers of Ly6C^+^ monocytes in draining lymph nodes (LNs), which were analyzed for their ability in capturing antigen and expressing CXCL9 and CXCL10; however, the ability of Ly6C^−^ Mos induced by CVC1302 to influence immune responses was not analyzed. This study aims to clarify the different temporal and spatial roles of Ly6C^+^ Mos and Ly6C^−^ Mos recruited by CVC1302 in initiating and modulating the immune response induced by vaccines. In conclusion, even both Ly6C^+^ Mo and Ly6C^−^ Mo had the potential to capture antigen and producing CXCL9 and CXCL10; the percentage of Ly6C^+^ Mo in OVA^+^ cells is much higher than Ly6C^−^ Mo, and the expression levels of CXCL9 and CXCL10 of Ly6C^+^ Mo were much higher when compared with Ly6C^−^ Mo. Our findings shed light on the development of new immunopotentiators and vaccines.

## 2. Materials and Methods

### 2.1. Mice

One-hundred-fifty-four BALB/c mice (five weeks old) were purchased from the college of veterinary medicine Yang Zhou University (Institute of comparative medicine) (Yangzhou, China). Mice were immunized after two days in order to minimize stress, which derived from transportation and environment change.

### 2.2. Antigen, Adjuvant, and Immunization

Ovalbumin (OVA) and Fluorescein OVA (OVAF) were purchased from Solarbio life sciences (Beijing, China). CVC1302 was prepared as the aqueous phase according to previous study [16]. OVA-ISA206, OVAF-ISA206, OVA-CVC1302-ISA206 and OVAF-CVC1302-ISA206 were prepared according to a previous study [17]. CVC1302 was mixed thoroughly with OVA or OVAF, then emulsified with ISA206. OVA or OVAF alone was emulsified with ISA206.

For analyzing the numbers of total Ly6C^+^ Mos and OVA^+^ Ly6C^+^ Mos, as well as total Ly6C^−^ Mos and OVA^+^ Ly6C^−^ Mos, twenty-four mice (twelve mice in each group) were immunized intramuscularly with OVAF-CVC1302-ISA206 or OVAF-ISA206, 50 μg OVAF per mice. Then, three mice from each group were sacrificed at 1, 3, 5 and 7 days post-immunization (dpi) to obtain popliteal LN, respectively.

For observing the distribution pattern of Ly6C^+^ Mos and Ly6C^−^ Mos, three mice were immunized intramuscularly with OVA-CVC1302-ISA206, 50 μg OVA per mice, and sacrificed at 1 dpi to obtain popliteal LN.

For clarifying the origins of Ly6C^+^ Mos and Ly6C^−^ Mos which infiltrated into the draining LNs, twenty-four mice (twelve mice in each group) were immunized in the ear pinnae with OVA-CVC1302-ISA206 or OVA-ISA206, 50 μg OVA per mice, and the ears were cut at 2 h after immunization. Three mice from each group were sacrificed at 1, 3, 5 and 7 dpi to obtain LN, respectively.

For analyzing the ability of Ly6C^+^ Mo and Ly6C^−^ Mo in producing CXCL9 and CXCL10, twelve mice (six mice in each group) were immunized intramuscularly with OVA-CVC1302-ISA206 or OVA-ISA206, 50 μg OVA per mice, and sacrificed at 3 dpi to obtain popliteal LN.

For observing the distribution pattern of CXCL9 or CXCL10-producing cells, two mice were immunized intramuscularly with OVA-CVC1302-ISA206 or OVA-ISA206, respectively, 50 μg OVA per mice, and sacrificed at 3 dpi to obtain popliteal LN.

For observing whether there was a difference in the distribution of OVA^+^ CD4^+^ T cells in mice immunized in the ear pinnae with OVA-CVC1302-ISA206 with or without ear-cut at 2 h after immunization, six mice were immunized intramuscularly with OVA-CVC1302-ISA206, 50 μg OVA per mice, and sacrificed at 14 dpi to obtain popliteal LN. OVA^+^ CD4^+^ T cells were sorted following stained with CD4-FITC and I-Ab OVA323-339 tetramer-APC and adoptively transferred into BABL/c recipients (5 × 10^5^ cells per mice) 1 d prior to immunization. FTY720 was administrated intraperitoneally (i.p.) at 0 h and 48 h, and LNs were harvested 72 h later to analyze the distribution of OVA-specific CD4^+^ T cells in LNs.

### 2.3. Preparation of Single-Cell Suspensions from Draining LNs

Popliteal LNs were sampled from mice and grounded gently in the 70 μm cell strainer to release LN cells. Single lymphocytes were washed twice with PBS containing 2% FBS. Lymphocytes were suspended in PRMI 1640 plus 10% fetal bovine serum (FBS) and adjusted to 2 × 10^7^ cells/mL.

### 2.4. FLOW Cytometry

For characterization of OVA^+^ Ly6C^+^ Mos and OVA^+^ Ly6C^−^ Mos, approximately 2 × 10^6^ cells per sample were blocked with mouse CD32/16 antibody (Miltenyi Biotec, Shanghai, China, 130-092-575), then stained with PE-conjugated anti-CX3CR1(ebioscience, Shanghai, China, 12-6099-42), PERCP-CY5.5-conjugated anti-CD11b (ebioscience, Shanghai, China, 45-0112-82), and APC-conjugated anti-Ly6C (ebioscience, Shanghai, China, 17-5932-82) at 4 °C for 30 min in the dark. Cells were washed twice with PBS (containing 2% FBS) and acquired by BD Accuri C6 flow cytometry. Cells were first identified as singlets via standard FSC and SSC gating, OVA^+^ cells were gated as OVA-FITC. The OVA^+^ Ly6C^+^ Mo was identified as OVA^+^ CD11b^+^ Ly6C^+^ CX3CR1^lo^, and the OVA^+^ Ly6C^−^ Mo was identified as OVA^+^ CD11b^+^ Ly6C^−^ CX3CR1^hi^.

For characterization of CXCL9- or CXCL10-producing Ly6C^+^ Mo and Ly6C^−^ Mo, approximately 2 × 10^6^ cells per sample were stained with PE-conjugated anti-CX3CR1, FITC-conjugated anti-CD11b (ebioscience, Shanghai, China, 11-0112-82), and PERCP-CY5.5-conjugated anti-Ly6C (ebioscience, Shanghai, China, 45-5932-82) at 4 °C for 30 min in the dark after being blocked with mouse CD32/16 antibody. Cells were fixed and permeabilized by using intracellular fixation and permeabilization buffer (ebioscience, Shanghai, China, 00-5523-00) according to the protocols, and then stained with CXCL9 recombinant rabbit monoclonal antibody (Invitrogen, 701117) or CXCL10 recombinant rabbit monoclonal antibody (Invitrogen, Shanghai, China, 701225) followed by donkey anti-rabbit IgG-AF647 (Invitrogen, Shanghai, China, A-31573). After washed with PBS (containing 2% FBS), cells were acquired by BD Accuri C6 flow cytometry. Cells were first identified as singlets via standard FSC and SSC gating, CXCL9 or CXCL10-producing Ly6C^+^ Mo were identified as CD11b^+^ Ly6c^+^ CX3CR1^lo^ CXCL9^+^/CXCL10^+^, and CXCL9- or CXCL10-producing Ly6C^−^ Mo were identified as CD11b^+^ Ly6C^−^ CX3CR1^+^ CXCL9^+^/CXCL10^+^.

Data were analyzed with FlowJo version 10.8.1 software (https://www.flowjo.com/solutions/flowjo/downloads, accessed on 5 June 2023).

### 2.5. Confocal Microscopy

Cryosections were prepared according to previous study [18]. For analyzing the distribution of Ly6C^+^ Mo and Ly6C^−^ Mo, the cryosections were incubated with Ly6C monoclonal antibody (Invitrogen, Shanghai, China, MA1-81899) and CX3CR1 recombinant rabbit monoclonal antibody (Invitrogen, Shanghai, China, 702321). For analyzing the distribution of OVA^+^ CD4^+^ T cells in draining LNs, the cryosections were incubated with 4′,6-diamidino-2-phenylindole (DAPI) (Thermo Scientific, Shanghai, China, 62248). For analyzing the distribution of CXCL9 and CXCL10-producing cells, the cryosections were incubated with CXCL9 recombinant rabbit monoclonal antibody and CXCL10 recombinant rabbit monoclonal antibody. After being washed twice with PBS containing 2% FBS, the images were acquired using Zeiss LSM image browser software version 4.2 (Zeiss, Carl Zeiss, Oberkochen, Germany).

### 2.6. Statistical Analysis

Statistical analysis was performed using GraphPad Prism version 5 (GraphPad Software, San Diego, CA, USA). Differences among groups were assessed using one-way analysis of variance followed by Tukey’s post hoc *t*-test. Differences between groups were assessed using a Student’s *t*-test. Values of *p* < 0.05 were considered statistically significant. All data shown in the manuscript are expressed as the means ± standard error of the mean (SEM).

## 3. Results

### 3.1. Recruitment of Ly6C^+^ Mo and Ly6C^−^ Mo in the Draining Inguinal LNs

To analyze the ability of CVC1302 in recruiting Ly6C^+^ Mo and Ly6C^−^ Mo, draining LNs were harvested from mice immunized with OVAF-CVC1302-ISA206 or OVAF-ISA206 at 1, 3, 5 and 7 dpi. Using specific antibodies recognizing cell type-specific markers, we detected the numbers of total Ly6C^+^ Mo and OVA^+^ Ly6C^+^ Mo (CD11b^+^ Ly6C^+^ CX3CR1^lo^), as well as total Ly6C^−^ Mo and OVA^+^ Ly6C^−^ Mo (CD11b^+^ Ly6C^−^ CX3CR1^hi^). As shown in Figure 1B, the numbers of both total Ly6C^+^ Mo and total Ly6C^−^ Mo in groups of mice immunized with OVAF-CVC1302-ISA206 or OVAF-ISA206 were increased to the peak at 3 dpi, and significantly higher numbers of total Ly6C^+^ Mo and total Ly6C^−^ Mo were observed in mice immunized with OVAF-CVC1302-ISA206 than in those immunized with OVAF-ISA206 at 3 dpi (*p* < 0.05 and *p* < 0.01). Even though the numbers of total Ly6C^+^ Mo and total Ly6C^−^ Mo were decreased from 5 dpi, there was also a significant difference in the numbers of total Ly6C^+^ Mo and total Ly6C^−^ Mo between the groups of mice immunized with OVAF-CVC1302-ISA206 or OVAF-ISA206 at 5 dpi. As shown in Figure 1C, the numbers of both OVA^+^ Ly6C^+^ Mo and OVA^+^ Ly6C^−^ Mo in groups of mice immunized with OVAF-CVC1302-ISA206 or OVAF-ISA206 were increased to the peak at 3 dpi, which was in line with the trends of total Ly6C^+^ Mo and total Ly6C^−^ Mo. Furthermore, the numbers of OVA^+^ Ly6C^+^ Mo and OVA^+^ Ly6C^−^ Mo in group of mice immunized with OVAF-CVC1302-ISA206 were significantly higher than those in group of mice immunized with OVAF-ISA206 (*p* < 0.01 and *p* < 0.001) at 3 dpi. However, unlike total Ly6C^+^ Mo and total Ly6C^−^ Mo, the numbers of OVA^+^ Ly6C^+^ Mo and OVA^+^ Ly6C^−^ Mo in group of mice immunized with OVAF-CVC1302-ISA206 were also significantly higher than those in group of mice immunized with OVAF-ISA206 (*p* < 0.01, *p* < 0.001 and *p* < 0.05) at 5 and 7 dpi. As shown in Figure 1D, the Ly6C^+^ Mo was the preferred subset to capture OVA-FITC among Ly6C^+^ Mo and Ly6C^−^ Mo in both groups of mice immunized with OVAF-CVC1302-ISA206 or OVAF-ISA206.

### 3.2. The Distribution of Ly6C^+^ Mo and Ly6C^−^ Mo Induced by CVC1302

In our previous study, we observed that CCR2^+^ Mo were distributed in the interfollicular region (IFR) and T zones in the draining LNs (under review). In this study, we aimed to clarify the function of Ly6C^+^ and Ly6C^−^ Mo recruited by CVC1302, so we analyzed the distribution of Ly6C^+^ Mo and Ly6C^−^ Mo in draining LNs. As shown in Figure 2, Ly6C^+^ Mo (red) and Ly6C^−^ Mo (green) have a different distribution pattern in draining LN sampled from mice immunized with OVA-CVC1302-ISA2-6. Ly6C^+^ Mo mainly distributed in the IFR and T zones, however, Ly6C^−^ Mo mainly distributed in subcapsular sinus (SCS) and IFR.

### 3.3. The Resource of Ly6C^+^ Mo and Ly6C^−^ Mo Recruited by CVC1302

It is known that the translocation of Mos into draining LNs could have occurred via two distinct routes: Mos are recruited into the inflamed sites of immunization, then migrate into draining LNs via the afferent lymphatics or Mos enter into draining LNs from blood via HEVs. In order to clarify the routes of Mo transporting into draining LNs, mice were immunized with OVA-CVC1302-ISA206 in the ear, and the site of immunization was surgically removed 2 h later. As shown in Figure 3, even though there was no significant difference in the numbers of Ly6C^+^ Mo at 1 dpi, there were significant reductions in the numbers of Ly6C^+^ Mo in the group of mice with ear removed at 3, 5 and 7 dpi (*p* < 0.0001); however, there was no significant difference in the numbers of Ly6C^−^ Mo between the groups of mice with or without ear removed at 1, 3, 5 and 7 dpi. The results demonstrated that the Ly6C^−^ Mo in draining LNs was mainly from the blood, whereas, the Ly6C^+^ Mo in draining LNs was mainly from the injection sites.

### 3.4. Ly6C^−^ MO Are a Prominent Source of CXCL9 and CXCL10

As known that CXCR3 ligands, CXCL9 and CXCL10, can attract CD4^+^ T cells into the IFR and medullary regions of the LN for optimal Th1 differentiation [19]. In our previous study, we found that CXCL9 and CXCL10 were mainly secreted by Mo (under review); in this study, we compared the potential of the Ly6C^+^ Mo and Ly6C^−^ Mo in secreting CXCL9 and CXCL10. As shown in Figure 4A,B, both Ly6C^+^ Mo and Ly6C^−^ Mo expressed the chemokines CXCL9 and CXCL10, but the mean fluorescence intensity (MFI) of CXCL9 and CXCL10 in Ly6C^−^ Mo were significantly higher compared with Ly6C^+^ Mo (*p* < 0.001 and *p* < 0.05). As shown in Figure 4C, CXCL9-producing cells and CXCL10-producing cells were mainly distributed in the SCS and IFR, which was line in the distribution of Ly6C^−^ Mo (as shown in Figure 2). Above all, we confirmed that CXCL9 and CXCL10 were mainly secreted by Ly6C^−^ Mo, which was attracted by CVC1302 into draining LNs.

### 3.5. The Distribution of Antigen-Specific CD4^+^ T Cells

In our previous study, OVA^+^ CD4^+^ T cells were observed that mainly distribute in IFR and T zones (under review), so we wonder whether the distribution of CD4^+^ T cells in the draining LNs had a direct relationship with the Ly6C^+^ Mo or Ly6C^−^ Mo or not. From the above results, we revealed that the number of Ly6C^+^ Mo decreased remarkably in mice with an ear-cut at 2 h after immunization. Here, we utilized confocal microscopy to observe whether there also had a change in the distribution of antigen-specific CD4^+^ T cells when the ears of immunized mice were removed. As shown in Figure 5, there was no remarkable change in the distribution of OVA^+^ CD4^+^ T cells between the two groups of mice with or without an ear-cut. It was demonstrated that the distribution of antigen-specific CD4^+^ T cells depended on the distribution of CXCL9 and CXCL10. Even though there was a significant decrease in the numbers of Ly6C^+^ Mo in mice after ear-cut, the distribution of OVA^+^ CD4^+^ T cells had no change in mice with ear-cut, so we concluded that the distribution of OVA^+^ CD4^+^ T cells mainly depended on the CXCL9 and CXCL10, produced by the Ly6C^−^ Mo, whose number had no change in mice with an ear-cut. As it is known that the distribution of CD4^+^ T cells in the T zones is mainly due to the chemokine CXCL19/CXCL21, produced by stromal cells, as the expression of CCR7 on the CD4^+^ T cells, as the ear-cut did not influent the stromal cells in the lymph nodes, after ear-cut, OVA^+^ CD4^+^ T cells also distributed into the T zones.

## 4. Discussion

CD4^+^ T cells are pivotal for the regulation of adaptive immune responses [20]. Native CD4^+^ T cells, after antigenic stimulation, differentiated into specific effector cells, such as Th1, Th2, Th17, follicular helper T cell (Tfh), induced T regulatory cell (iTreg), regulatory type 1 cell (Tr1) and Th9 [21,22]. Th1 cells play important roles in the elimination of intracellular pathogens, mainly secreting IFN-γ, TNF-α and IL-2. It believed that Th1 cells coordinated humoral immunity and cellular immunity through the production of the cytokines [18,23].

Mos divided into Ly6C^+^ Mos and Ly6C^−^ Mos, which transferred into immune sites dependent on different receptors CCR2 and CX3CR1, respectively. And in different immune microenvironments in draining LNs, Ly6C^+^ Mos and Ly6C^−^ Mos mediated immune responses in different ways. As reported that CCR2^+^ Mos infiltrate the LN into IFR and T zone from the blood in response to LPS, TLR4 agonist, formulated with IFA, which induces CD4^+^ T cells localization into the IFR by releasing CXCL10 [19]. It was demonstrated that CpG, the TLR9 agonist, promoted Tfh differentiation due to IL-6 secreted by Ly6C^+^ Mo-derived dendritic cells [24].

It was demonstrated that the expression of CXCL9 and CXCL10 by Mo depended on IFN-α and IFN-β. We have confirmed that CVC1302 could induced significantly enhanced IFN-α and IFN-β in draining LNs from mice immunized with CVC1302-adjuvanted FMDV killed vaccine [18].

We discovered that CVC1302 could enhanced the differentiation of IFN-γ^+^ Th1 cells and the levels of antigen-specific IgG2a antibody [16,25]. Hence, in this research, we paid attention on the definition of the ability of CVC1302 in mediating the functions of Ly6C^+^ Mo and Ly6C^−^ Mo to elicit robust Th1 responses in mice. From the results, we found that Ly6C^+^ Mo transferred into draining LNs mainly from injection sites, whereas Ly6C^−^ Mo infiltrated into draining LNs mainly from blood, which was demonstrated by the ear-cut experiment. With the results of confocal microscopy, we also found that Ly6C^+^ Mo and Ly6C^−^ Mo had a different pattern in the distribution in draining LN respond to CVC1302. As known that IFR was a region between B cell zone and T cell zone, when antigen-specific CD4^+^ T cells were attracted into IFR, which facilitated the activation of antigen-specific B cells. In this study, we found that even though both Ly6C^+^ Mo and Ly6C^−^ Mo infiltrated into IFR, there was a significant difference in the expression levels of CXCL9 and CXCL10 between Ly6C^+^ Mo and Ly6C^−^ Mo. Hence, we concluded that Ly6C^+^ Mos are a prominent cell subset to capture antigens when compared with Ly6C^−^ Mos, which are prominent a prominent way to produce CXCL9 and CXCL10 to attract antigen-specific CD4^+^ T cells into IFR, and encounter IL-12^+^ DCs to differentiate into Th1. Lian et al. demonstrated that LPS formulated with IFA recruited CCR2^+^ Mo into the IFR and T zones, where they produced CXCL10 and attract CD4^+^ T cells, in our study, we utilized specific fluorescent antibodies to further clarify the subtype of recruited Mo and their respective potentials in inducing immune responses.

In our previous study, we demonstrated that CVC1302-adjuvanted FMDV killed vaccines inducing higher percentage of germinal center (GC) B cells, which differentiated into long-lived plasma cells (LLPCs) to provide long-term humoral immunity, and relied on the differentiation of Tfh cells. Based on the results derived from the research, we clarified that CVC1302 inducing promoted humoral immunity and also depended on enhanced Th1 cells, which were attracted into the IFR.

## 5. Conclusions

In this study, we demonstrated the CVC1302-mediated enhancement of monocyte recruitment. However, the different subtypes of Mo had different potentials in inducing immune responses. The Ly6C^+^ Mos infiltrated into draining LNs from injection sites, and Ly6C^−^ Mos infiltrated into draining LNs from blood. Furthermore, Ly6C^+^ Mos distributed into IFR and T zones, and Ly6C^−^ Mos distributed into SCS and IFR. Meanwhile, Ly6C^+^ Mos play an important role in initiating the immune responses by capturing antigens, and Ly6C^−^ Mos play an important role in facilitating the differentiation of Th1 by producing CXCL9 and CXCL10. The results from the research shed light on the development of new immunopotentiators to establish humoral and cellular immunity.

## Figures and Tables

**Figure 1 vaccines-12-00086-f001:**
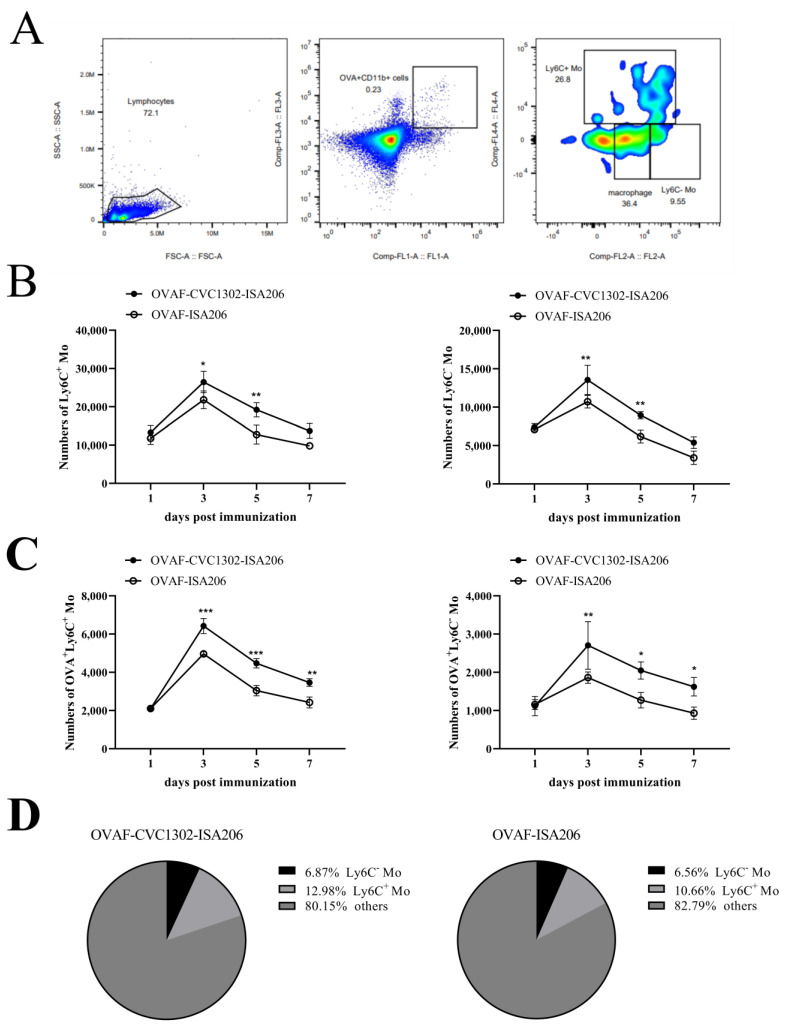
Kinetics of Ly6C^+^ Mo and Ly6C^−^ Mo recruitment into the draining LNs in responses to immunization with OVAF-CVC1302-ISA206 or OVAF-ISA206. BALB/c mice (twelve mice in each group) were immunized with OVAF-ISA206 or OVAF-CVC1302-ISA206. Popliteal LNs were harvested at 1, 3, 5 and 7 dpi and the numbers of targeted cells were evaluated with flow cytometry. (**A**) Representative flow cytometry plot showing OVA^+^ Ly6C^+^ Mo and OVA^+^ Ly6C^−^ Mo. (**B**) The numbers of total Ly6C^+^ Mo (**left**) and total Ly6C^−^ Mo (**right**) in draining LNs at 1, 3, 5 and 7 dpi. (**C**) The numbers of OVA^+^ Ly6C^+^ Mo (**left**) and OVA^+^ Ly6C^−^ Mo (**right**) in draining LNs at 1, 3, 5 and 7 dpi. (**D**) The percentages of OVA^+^ Ly6C^+^ Mo and OVA^+^ Ly6C^−^ Mo in OVA^+^ cells in mice immunized with OVAF-CVC1302-ISA206 (**left**) and OVAF-ISA206 (**right**). Data are presented as mean ± SEM; * *p* ≤ 0.05, ** *p* ≤ 0.01, *** *p* ≤ 0.0001.

**Figure 2 vaccines-12-00086-f002:**
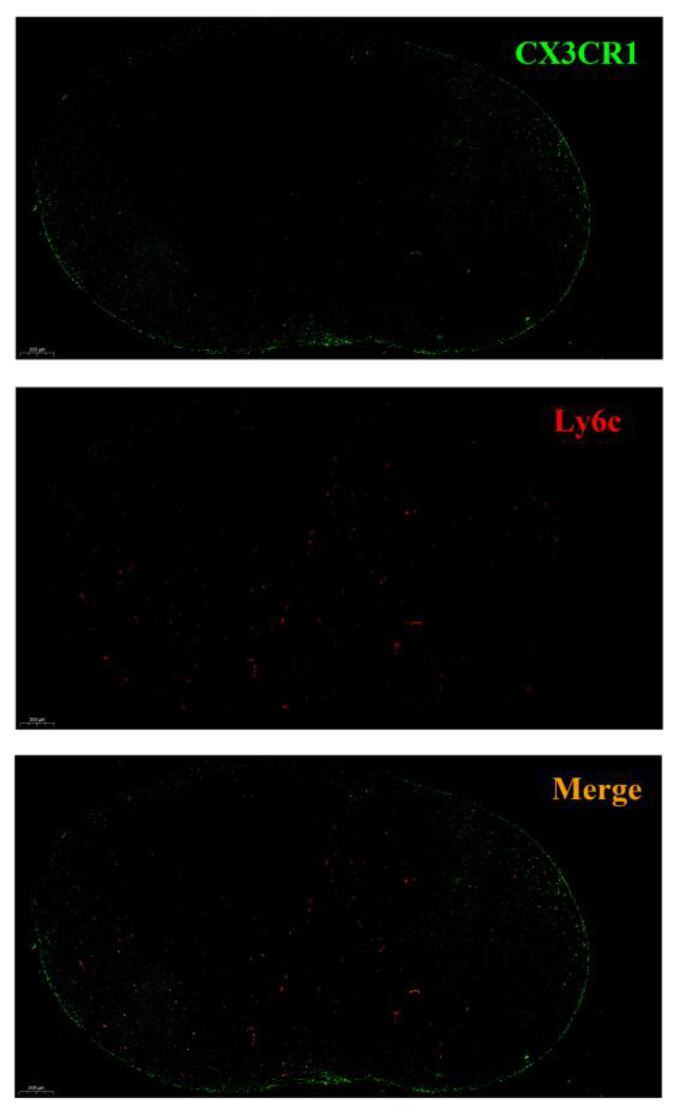
The distribution pattern of Ly6C^+^ Mo and Ly6C^−^ Mo in draining LNs in response to immunization with OVA-CVC1302-ISA206. Mice were immunized intramuscularly with OVA-CVC1302-ISA206 and LNs were sampled at 3 dpi to analyze the distribution of Ly6C^+^ Mo and Ly6C^−^ Mo by confocal microscopy. LNs were stained with anti-Ly6C (red) and anti-CX3CR1 (green).

**Figure 3 vaccines-12-00086-f003:**
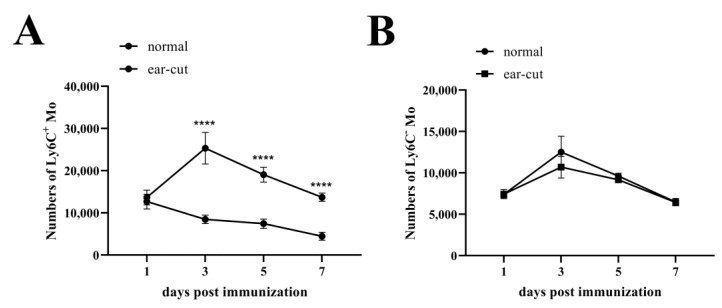
The origins of Ly6C^+^ Mo and Ly6C^−^ Mo infiltrated into the draining LNs in response to immunization with OVA-CVC1302-ISA206. BALB/c mice (twelve mice in each group) were immunized in the ear with OVA-CVC1302-ISA206, one group of mice were ear excised 2 h after immunization. Popliteal LNs were harvested at 1, 3, 5 and 7 dpi and stained with specific fluorescent antibody, then analyzed by flow cytometry. The numbers of total Ly6C^+^ Mo (**A**) and total Ly6C^−^ Mo (**B**) at the indicated time-points post-immunization. Data are presented as mean ± SEM; **** *p* ≤ 0.0001.

**Figure 4 vaccines-12-00086-f004:**
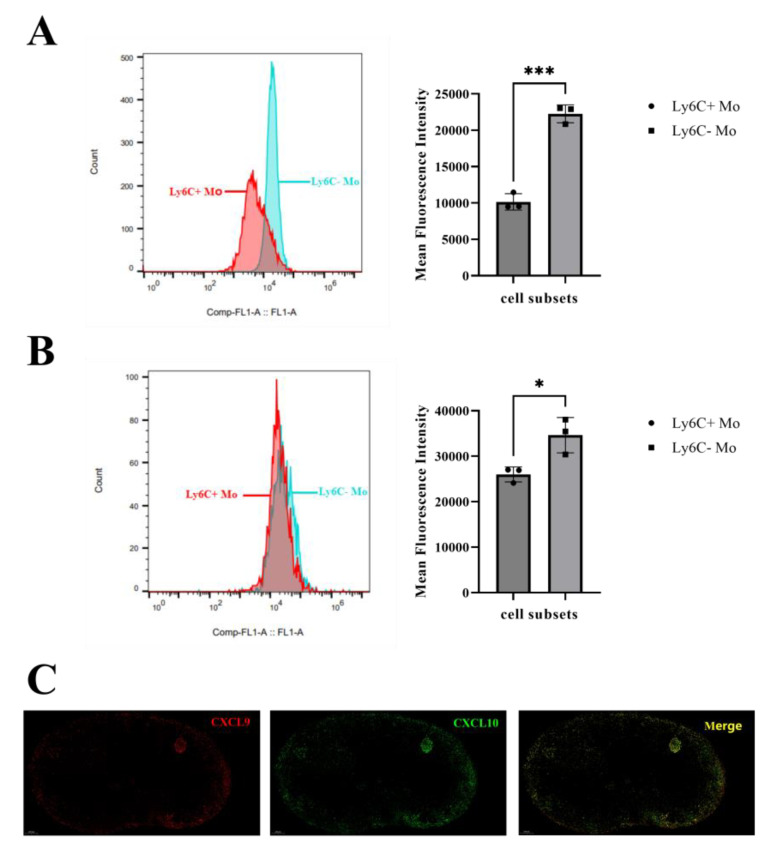
The potential of Ly6C^+^ Mo and Ly6C^−^ Mo in releasing CXCL9 and CXCL10 in response to immunization with OVA-CVC1302-ISA206. LNs were sampled from mice, immunized with OVA-CVC1302-ISA206, at 3 dpi to analyze the percentage of CXCL9 or CXCL10-producing Ly6C^+^ Mo and Ly6C^−^ Mo. (**A**) (**left**) Representative plot of CXCL9^+^ cells at 3 dpi. (**right**) MFI of CXCL9 expression with Ly6C^+^ Mo and Ly6C^−^ Mo. (**B**) (**left**) Representative plot of CXCL10^+^ cells at 3 dpi. (**right**) MFI of CXCL10 expression with Ly6C^+^ Mo and Ly6C^−^ Mo. (**C**) The distribution of CXCL9 or CXCL10-producing cells. Popliteal LNs were harvested at 3 dpi from mice immunized with OVA-CVC1302-ISA206. Cryosections (6 μm) of LNs were stained with anti-CXCL9 and anti-CXCL10, then visualized using confocal microscopy. Data are presented as mean ± SEM; * *p* ≤ 0.05, *** *p* ≤ 0.0001.

**Figure 5 vaccines-12-00086-f005:**
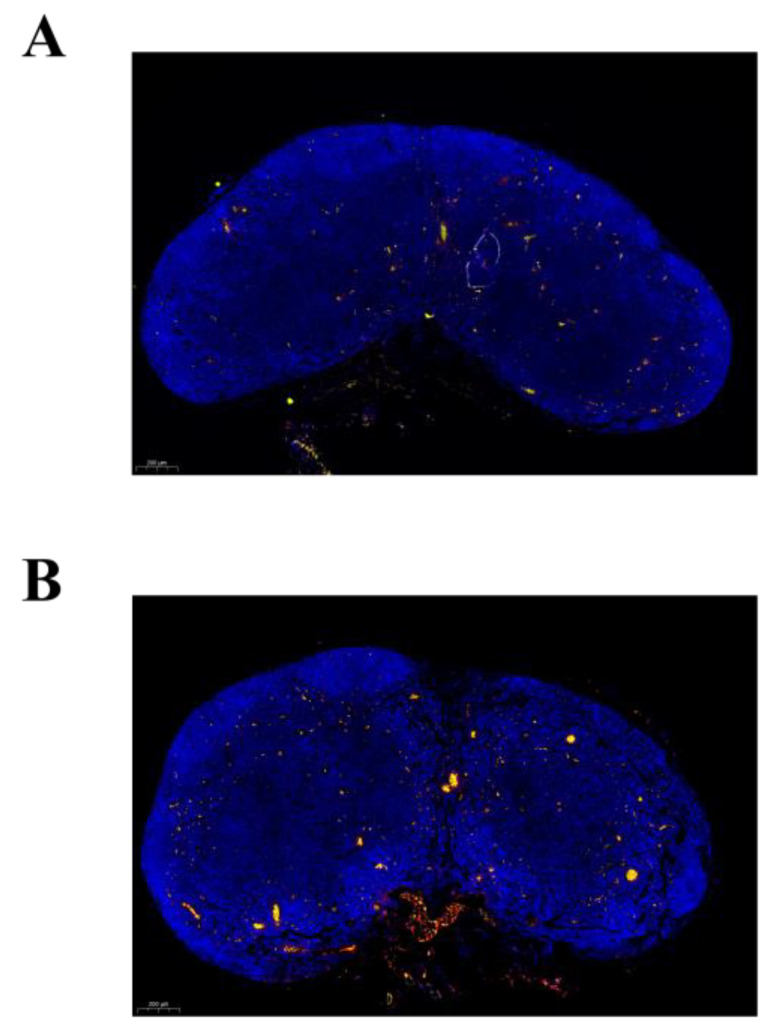
The distribution of antigen-specific CD4^+^ T cells in response to immunization with OVA-CVC1302-ISA206. LNs were harvested at 14 dpi from mice immunized with OVA-CVC1302-ISA206, lymphocytes were stained with CD4-FITC and I-Ab OVA323-339 tetramer-APC, then OVA^+^ CD4^+^ T cells were adoptively transferred into recipient mice 1 day prior to immunization. FTY720 was administrated intraperitoneally (i.p.) at 0 h and 48 h, and lymph nodes were harvested 72 h later. (**A**) Mice were immunized in the ear, which was excised 1 h after immunization. Representative confocal images of the localization of transferred CD4^+^ T cell (yellow) to lymph node niches. (**B**) Mice were immunized in the ear. Representative confocal images of the localization of transferred CD4^+^ T cell (yellow) to lymph node niches.

## Data Availability

The data presented in this study are available in this article.

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
