# Peer review of "Analysis of CVC1302-Mediated Enhancement of Monocyte Recruitment in Inducing Immune Responses"

_vaccines, 2024, doi:10.3390/vaccines12010086_

Round 1

Reviewer 1 Report

Comments and Suggestions for Authors

In this study, the authors aim to investigate how CVC1302, a complex of PRRs agonists, influences the recruitment of both Ly6C+ monocytes (Mo) and Ly6C- Mo in initiating and modulating the immune response induced by OVA-FITC vaccines. The examination involves comparing two immunization groups, one with CVC1302 and one without, and exploring different administration routes, including the use of ear-cutting to eliminate local immunization spots post-administration.

The findings reveal that although both Ly6C+ Mo and Ly6C- Mo have the potential to capture antigens, they are recruited into draining lymph nodes through distinct pathways, displaying different distribution patterns within the local lymph nodes. Ly6C+ Mo are observed to distribute into the IFR and T zones, while Ly6C- Mo preferentially distribute into the SCS and IFR. Additionally, Ly6C+ Mo emerge as a more prominent cell subset for antigen capture but exhibit lower efficiency in expressing CXCL9 and CXCL10 and are relatively less effective in recruiting antigen-specific CD4+ T cells into draining lymph nodes compared to Ly6C- Mo.

Building on their previous study, the authors conclude that in response to CVC1302, Ly6C+ Mo migrates into lymph nodes from the injection sites, while Ly6C- Mo originates from the blood. They localize to different zones of lymph nodes and perform distinctive functions, with Ly6C+ Mo aiding in antigen capture and Ly6C- Mo facilitating the differentiation of Th1 by producing CXCL9 and CXCL10. 

These key findings does help enhance our understanding of vaccine immunology. The experiments are designed to maintain a clear study focus, and the data is presented with reasonable precision. While the overall quality of the study is commendable, there is room for improvement in the writing. I am inclined to accept this manuscript with appropriate revisions.

I have a few questions listed below:

Is there any in vitro data to show that Ly6C+ monocytes (Mo) have a higher capacity to capture OVA-FITC antigen compared to Ly6C- Mo? Do the authors have some explanation why these two Mo subsets are recruited into draining lymph nodes through different routes?

The authors mentioned in the previous study that OVA+ CD4+ T cells mainly distributed in the IFR and T zones, which is more consistent with the distribution pattern for Ly6C+ Mo than Ly6C- Mo. However, this study demonstrated no remarkable change in the distribution of OVA+ CD4+ T cells between the two groups of mice with or without ear-cut. Does this imply that the CXCL9 and CXCL10 produced by Ly6C+ Mo have no contribution to the antigen-specific CD4+ T cell recruitment at all?

In the study, the authors also demonstrated that the distribution of antigen-specific CD4+ T cells depended on the distribution of CXCL9 and CXCL10. Is there any good explanation for the recruitment of OVA+CD4+ T cells to T cell zones with ear-cut post immunization?

Have the authors ever attempted to quantify the recruitment of antigen-specific CD4+ T cells with and without ear-cut?

Comments on the Quality of English Language

While the overall quality of the manuscript is commendable, there is room for improvement in the writing, and it could benefit from minor polishing

Reviewer 2 Report

Comments and Suggestions for Authors

The presented manuscript compares monocyte recruitment during an immune response with and without CVC1320 adjuvant and demonstrates an effect of nearly 1.5-fold increase in monocyte numbers in lymph nodes. However, subsequent experiments were performed only on mice immunized with CVC1320 without comparison, and the observed effects were attributed solely to this adjuvant.

In my opinion, the conclusions and title of the paper should be revised to “...CVC1320-mediated enhancement of monocyte recruitment...” instead of “...monocytes recruited by CVC1320...”.

The abundance of undeciphered or unexplained abbreviations makes the work difficult to understand for non-specialists.

L55, 56 The single use of "reactive oxygen species" makes the acronym ROS unnecessary. SLAN needs to be deciphered for a wider readership. The same goes for OVA and IFR (at first usage), GM-CSF, Flt3L, FBS, PE, APC, PERCP, DAPI, IFA.

A flowchart/table of study design including immunization groups, number of animals, immunogens, time of sacrifice, study material, identifiable markers, and purpose of each experiment would be very helpful for overall understanding of the work.

Some minor points are following.

L68. Please, characterize what is CVC1302. Is it immunopotentiator, adjuvant mixture or b-glucan. Provide a citation.

L88. Please, indicate if OVA-ISA206, OVAF-ISA206 etc. are emulsions. Specify OVA dose per animal. 

L 91. Detail immunization groups, immunization scheme. How many mice were immunized and with what immunogen?

L101-124. The source of antibodies and conjugates should be noted.  For ease of understanding of the experiments, I recommend that authors organize a section on “Immunostaining” in the Materials and Methods where the identification of each cellular marker will be described.

L103, 153. Which mice were used to obtain inguinal LN ?

L 108. Specify that BD Accuri C6 is flow cytometer.

L119, 120 Indicate if CXCL9 or CXCL10 monoclonal antibody are rabbit antibody since donkey anti-rabbit IgG-AF647 were used.

L132 How many selected OVA+ CD4+ T cells were transferred into BABL/c recipients (n=?).

Fig. 1 The quality of the images is very low, so the inscriptions are not readable

Discussion. The novelty of the present study/approaches should be noted and the new results obtained should be presented in comparison with other published studies.

Round 2

Reviewer 2 Report

Comments and Suggestions for Authors

The authors adderessed all the reviewer comments.

The requested explanations are provided.